# A Review on Motor Imagery with Transcranial Alternating Current Stimulation: Bridging Motor and Cognitive Welfare for Patient Rehabilitation

**DOI:** 10.3390/brainsci13111584

**Published:** 2023-11-12

**Authors:** Rosary Yuting Lim, Kai Keng Ang, Effie Chew, Cuntai Guan

**Affiliations:** 1Institute for Infocomm Research, Agency for Science Technology and Research, A*STAR, 1 Fusionopolis Way, #21-01 Connexis, Singapore 138632, Singapore; kkang@i2r.a-star.edu.sg; 2School of Computer Science and Engineering, Nanyang Technological University, 50 Nanyang Ave., #32 Block N4 #02a, Singapore 639798, Singapore; ctguan@ntu.edu.sg; 3Division of Rehabilitation Medicine, Department of Medicine, National University Hospital, 5 Lower Kent Ridge Rd, Singapore 119074, Singapore; effie_chew@nuhs.edu.sg; 4Yong Loo Lin School of Medicine, National University of Singapore, 10 Medical Dr, Singapore 117597, Singapore

**Keywords:** motor imagery, transcranial alternating current stimulation (tACS), transcranial direct current stimulation (tDCS), working memory, attention, fatigue, stroke rehabilitation

## Abstract

Research has shown the effectiveness of motor imagery in patient motor rehabilitation. Transcranial electrical stimulation has also demonstrated to improve patient motor and non-motor performance. However, mixed findings from motor imagery studies that involved transcranial electrical stimulation suggest that current experimental protocols can be further improved towards a unified design for consistent and effective results. This paper aims to review, with some clinical and neuroscientific findings from literature as support, studies of motor imagery coupled with different types of transcranial electrical stimulation and their experiments onhealthy and patient subjects. This review also includes the cognitive domains of working memory, attention, and fatigue, which are important for designing consistent and effective therapy protocols. Finally, we propose a theoretical all-inclusive framework that synergizes the three cognitive domains with motor imagery and transcranial electrical stimulation for patient rehabilitation, which holds promise of benefiting patients suffering from neuromuscular and cognitive disorders.

## 1. Introduction

Motor imagery (MI), generally defined as the mental act of imagining a given motor movement (e.g., upper or lower limb movements) without actuating its execution, has been neuroanatomically localized and functionally attributed to four main regions of the brain: posterior parietal cortex, premotor area (PMA), supplementary motor area (SMA), and primary motor area [1].

The posterior parietal cortex is implicated in translating visual inputs into motor execution information during the planning stage of movement [2,3]. It then relays this information to the PMA and SMA. PMA, anatomically located anterior to the primary motor area, has a role in the sensory guidance of movement. Connectivity analysis has shown that the PMA plays a central role in bridging MI with cognition- and other motor-related brain regions [4,5]. On the other hand, SMA, situated medial to PMA and anterior to primary motor area, is necessary for planning complex motor movements and facilitating the coordination of bimanual movements [6,7]. Both the PMA and SMA transmit information to the primary motor area and to brainstem motor regions for motor execution. In a nutshell, the sensory inputs enter via the posterior parietal cortex to be relayed to the PMA and SMA regions, before being transmitted to the primary motor area for motor execution.

The majority of stroke survivors do benefit from conventional physical motor therapy. However, a substantially significant improvement is observed when such rehabilitative therapies incorporate sessions of MI to augment physical therapy [8,9,10] for an enhanced outcome according to the Hebbian effect.

Most of the studies involving MI utilized electroencephalogram (EEG) or functional near infrared spectroscopy (fNIRS) to construct a Brain-Computer Interface (BCI) to assist the rehabilitation. Multimodal interventions in patient studies, comprising of a primary motor or cognitive task and a secondary treatment intervention, have elicited positive behavioral or performance outcomes when transcranial electrical stimulation (tES) is utilized as the secondary intervention in experimental paradigms [11,12,13,14]. Hence, it is imperative to first understand neurophysiological implications of certain types of tES to potentially induce longer lasting effects and improvements in a wider group of subjects with fewer exclusions.

Apart from suffering from motor impairments, stroke survivors live with other repercussions of stroke, such as cognitive impairments [15,16], and post-stroke depression [17].

Two major factors that likely impact a patient’s progress in rehabilitation are motivation and fatigue levels. Motivation is a component of affect that could impact on the degree of attention allocated to the task at hand. Certain tasks during the therapy may be unsuitable, which could lead to failed attempts resulting in a vicious cycle of negativity—an effect stemming from negative performance feedback on motivation. As a result, those feelings of dejection might negate the benefits of rehabilitation [18]. Fatigue can be subdivided into two main classes: active fatigue and passive fatigue. Active cognitive fatigue stems from cognitive overload, while passive cognitive fatigue stems from cognitive underload. Patients with multiple sclerosis, Parkinson’s Disease, or fibromyalgia suffer from the consequences of cognitive fatigue as well [19,20,21].

Regarding MI tasks designed for rehabilitation, it will be advantageous to integrate real-time monitoring of an individual’s level of attentiveness and fatigue into a protocol that incorporates the use of a suitable tES. Therefore, this paper aims to review and evaluate, using some examples from recent literature, the feasibility of integrating tES, specifically comparing between transcranial direct current stimulation (tDCS) and transcranial alternating current stimulation (tACS) and measures of motivation and fatigue in future MI-related protocols for neuromuscular and/or neurocognitive patient populations. At the end of the paper, this review will conclude with a suggested framework to explore how the mentioned cognitive domains can be integrated into a potential rehabilitation protocol design for future work.

## 2. Review Selection and Criteria

Specific to this review, the Preferred Reporting Items for Systematic Reviews and Meta-Analysis (PRISMA) guidelines were followed.

Published articles in English were surveyed from PubMed with the following search terms:

“Transcranial alternating current stimulation [attention/working memory/fatigue/motor imagery] adults”

“tACS [attention/working memory/fatigue/motor imagery] adults”

Literature from 2016 to 2023 in the following cognitive domains were investigated: attention, working memory, and fatigue—the three major areas of cognitive concerns pertaining to stroke patient rehabilitation. The inclusion criteria are such that the studies:

1. had applied tACS or tDCS to investigate at least one of the cognitive domains;

2. studies conducted in healthy young adults;

3. number of analyzed subjects greater than ten;

4. stimulate continuously for at least 5 min.

The search on PubMed conducted in May 2023 (Figure 1) for articles from 2016 to 2023 yielded 4 articles for “motor imagery”, 283 articles for “attention”, 263 articles for “working memory”, and none for “fatigue”. Upon removal of duplicates and transcranial direct current literature, which is irrelevant to this review, it yielded 33 articles for “attention”, 31 articles for “working memory”, and none for “fatigue”.

In the domain of “motor imagery”, all four articles are deemed relevant and are included for discussion.

In the domain of “attention”, removal of irrelevant articles with non-matching keywords (n = 14), reviews or meta-analyses or dataset reports (n = 5), patient studies and case reports (n = 4), and studies that did not meet one or more of the inclusion criteria (n = 1) resulted in 9 articles to be included.

In the domain of “working memory”, removal of irrelevant articles with non-matching keywords (n = 7), reviews or meta-analyses or dataset reports (n = 7), patient studies and case reports (n = 3), and studies that did not meet one or more of the inclusion criteria (n = 4) resulted in 10 articles to be included.

In the domain of “fatigue”, since there are no return results using the above keywords during the search, the search terms were modified as “EEG passive driving fatigue”. For this domain, the inclusion criteria are as follows:

1. investigations conducted with EEG;

2. studies conducted in healthy young adults;

3. number of analyzed subjects greater than ten;

4. investigation done on passive fatigue (i.e., driver’s fatigue).

The search yielded 7 results. With the removal of removal of irrelevant articles with non-matching keywords (n = 4) resulted in 3 articles to be included.

In accordance to the scope of this review, TMS-related literature is excluded from the following evaluation and discussion.

## 3. Transcranial Direct Current Stimulation (tDCS) and Transcranial Alternating Current Stimulation (tACS)

### 3.1. Working Principles and Mechanisms

There are two commonly used types of tES in the research field: tDCS and tACS. While both are non-invasive brain stimulation delivering exogenous current to the brain via surface electrodes, tDCS delivers a constant direct current while tACS delivers an alternating sinusoidal current. When applying tDCS, current is delivered through the anode, resulting in excitation (depolarization) of the cortical region beneath its placement. It is postulated to be a disinhibitory mechanism from reduced localized gamma-aminobutyric acid (GABA)—and is received at the cathode’s end—resulting in inhibition (hyperpolarization) of the cortical region beneath its placement via inhibition of glutamatergic activity [22]. Thus, tDCS is capable of altering the excitability of neuronal populations to influence their firing probability.

On the other hand, application of tACS induces the flow of current from the stimulating electrode (i.e., anode) to the reference electrode (i.e., cathode) in the first half cycle of the sinusoidal waveform before it flows in the reverse direction in the second half; thereby resulting in a bi-directional current flow between stimulating and reference electrodes. By inducing a rhythmic oscillating flow of exogenous current relative to the endogenous brain oscillation patterns, tACS is capable of both stimulating and influencing endogenous brain oscillations that are frequency specific. Such a phenomenon is termed “neuronal entrainment”, wherein cortical neurons located in the proximity of tACS influence will either resonate stochastically or collectively as a targeted population to the externally applied oscillating current with a greater probability to fire [23].

### 3.2. Implications on Neurophysiology

The human brain is home to about 100 billion neurons with an estimated of 200 trillion connections within the network, all of which are compacted into tissue matter the size of approximately 1300 cubic centimeters [24,25]. In addition, it is mandatory that these neurons are organized so that meaningful functional groups are appropriately spaced in terms of spatial coordinates as well as with the appropriate synaptic distances amongst other neuronal populations. Such distributions comprise of both short- and long-range connections from one neuron to another, stemming from one end of the brain to the other.

However, physical connections alone are not computationally optimal. Neurons are not perfect, and errors do occur along their axons as signals propagate. On top of which, there are also conduction delays. Another important element is the presence of an inhibitory network. Without inhibition, all neurons will undergo continuous excitatory outbursts given any stimulus, as with the case of epileptic patients [26]. The activity of inhibitory neurons also facilitates a rhythmic mixture of transient and prolonged synchronous excitatory activity observed as brain oscillations.

This is a postulation of how the brain coordinates various populations to process information swiftly and effectively. It strongly suggests that the physical connection of how neurons are wired together is not as important as to how various populations of neurons synchronize their activities to the rhythm from the inhibitory network. Hence, it is arguable that tACS, with the ability to entrain and alter intrinsic brain rhythms, is a value-added modality that may prove to be more effective and efficient than tDCS.

A coordinated voluntary MI or movement intention theoretically follows a sequential flow of events, where sensory information flows from the posterior parietal cortex to the PMA and SMA and then reaches the primary motor cortex for actuation of the muscle groups. However, it has been found that amongst these four brain subregions exist bi-directional anatomical projections [27]. Given that bi-directional, or reciprocal, projections exist in those four brain regions necessary for MI and motor execution, it may suggest that the use of tACS has an advantage over tDCS based on its capacity to synchronize communication between regions of interest in a bi-directional, frequency-specific manner and strengthen their functional connectivity.

Indeed, tDCS has been shown to influence neuronal firing rates, which potentially modulates long-term potentiation (LTP) of a brain region to provide good motor learning outcomes [11,12,13,14,28,29,30]. However, the observed transient behavioral improvements are likely attributed to a short-lived LTP that could not be maintained due to the lack of structural plasticity factors. It has been shown that maintenance of LTP would require the biosystem to possess resources for certain structural modifications at the neuron synapses, e.g., new protein synthesis [31], new dendritic spines formation, growth of existing spines and associated postsynaptic densities (PSDs) [32], and insertion and active maintenance of an increased pool of α-amino-3-hydroxy-5-methyl-4-isoxazole propionic acid receptors (AMPARs) relative to *N*-methyl-D-aspartate receptors (NMDARs) [33] leading to an increase in size of PSDs, which triggers an increase in size of the corresponding presynaptic active zone. Such structural plasticity factors, which are prerequisite to maintain LTP, are unlikely to be present in all subjects, especially patient populations, therefore rendering tDCS unlikely to provide longer lasting benefits in patient rehabilitation paradigms. One consequence of the loss of LTP in neuronal populations is an altered neuronal oscillation in the network. Animal studies have shown that the regular theta-gamma oscillations in subcortical brain regions such as the hippocampus are disrupted following a deficit in LTP, thereby resulting in an impairment in sensory information processing and loss of hippocampal function [34]. Nonetheless, it is evident that cortical connectivity reorganization occurs in individuals, even for stroke patients. Hence, despite impaired mobility or damaged brain structure(s), they still can practice MI on the premise that the brain remains functionally plastic (i.e., capable of reorganizing activity and connectivity of neuronal populations) [35,36]. On top of the ability of bypassing the structural plasticity prerequisites needed by tDCS, tACS has also been shown to cause long-range, indirect entrainment effects in central neurons by stimulating peripheral sensory neurons [37]. Hence, tACS may surface as the more appropriate stimulation modality in a MI experimental design to re-establish patterns of frequency-specific brain oscillations and facilitate region-specific cortico-cortical functional connectivity that closely mimics biological characteristics [38].

### 3.3. Potential Adverse Effects

A recent review article has collated a list of negative side effects experienced by subjects from the application of tDCS and tACS, respectively [39]. It has been pointed out that, while majority of the side effects reported were mild and transient, some tDCS subjects had experienced persistent adverse effects such as skin irritation, headache, mania/hypomania in certain patient groups, and a serious occurrence of seizure noted in one particular patient, likely with a history of epilepsy. For tACS subjects, reports include retinal phosphenes or visual field flashes, cutaneous sensations, and dizziness. Unlike that of tDCS, the adverse effects reported due to tACS were all relatively milder. Notwithstanding the relatively fewer tACS studies, the fact that declarations of adverse effects experienced by tACS subjects were temporary shows that tACS is a potentially more comfortable type of tES for users and patients alike to undergo. It is not within the scope of this paper to provide an extensive coverage of all adverse effects experienced for tDCS and tACS. Interested readers are advised to refer to the article reported by Matsumoto and Ugawa (2017) [39].

### 3.4. Motor Imagery-Related tACS Studies

Unlike motor execution, motor imagery can be defined as the act of mental activation or reactivation of a motor movement without an observable overt output. It is a cognitive process involved in motor planning that results in the activation of specific cortical regions such as the premotor cortex, supplementary motor area, posterior parietal cortex, and primary motor cortex.

To date, there have only been four studies, conducted from 2016 to date, that investigated the implications of tACS on motor imagery (Table 1). A similar conclusion deduced from all four studies is that the spectral power of both alpha and beta oscillatory bands were observed to decrease during motor imagery, an observation indicative of a disinhibitory mechanism in the motor planning phase prior to motor execution i.e., when executing motor imagery. In addition to this overarching finding, Brinkman et al. (2016) also showed that a delivery of 10 Hz tACS resulted in faster reaction times in subjects when multiple movements are required during the given task [40]. They rationalized that such facilitatory effects are attributed to the enhancement of alpha band power in the ipsilateral cortex to the task hand so that it increases inhibition of the other competing non-task hand. This interpretation was also emphasized in another study conducted in stroke patients by Naros and Gharabaghi (2016) [41], wherein the observations presented themselves in terms of changes in beta power event related desynchronization (ERD) with tACS. They showed a decrease in ipsilesional resting state beta activity as compared to the contralesional hemisphere, which suggests that beta-tACS has an effect at “readying” the subject during the resting phase for a pending planned action. The authors coined this phenomenon as increasing the “specificity of the classifier’s true negative rate”. Although it did not directly influence the patients’ MI-related beta-ERD, the fact that beta-tACS could facilitate better discriminatory performance classification in stroke patients is significant.

As highlighted by Naros and Gharabaghi (2016) [41], it seems like focusing on the time window that encapsulates the MI task and analyzing for oscillatory band power changes in terms of ERD is vital to reveal effects of tACS on MI itself. More recent works, such as that by Xie et al. (2021), conducted an in-depth comparative investigation between the effects of tDCS and tACS [30]. They found that although the relative amplitude and duration of MI-related ERD were enhanced and prolonged, respectively, only a specific time bin of 3–5 s of the MI epoch was truly useful in analyzing for MI-related neural activation and representation. In addition to the temporal specificity of analysis for MI, the newest publication by Zhang et al. (2023) demonstrated that phase specificity of tACS delivered to certain brain regions is also pivotal to the facilitate different difficulty levels of MI tasks [42]. They found that anti-phase tACS stimulation was able to significantly decrease alpha band power in the motor cortex region during MI tasks, especially that of the complex written MI task. It was reasoned that since MI is associated with decrease in band power as seen in regions of ERD mainly in the motor cortex region, in-phase tACS was likely associated with an increase in oscillatory power in the stimulated frequency. Therefore, given their results, it would be counterintuitive to the motivation of facilitating MI effects should in-phase tACS be utilized.

Taken together, the overall research direction set for tACS in MI-related work seems to be heading towards an understanding of the basic underlying mechanisms of how tACS is modulating changes in the brain. From prior research, we gathered that ERD occurs in contra-hemispheric regions where there is reduced oscillatory band power during MI tasks, and that such reductions are often seen in the alpha and beta power bands. Hence, the future questions for related work should be: what are the specific frequencies to stimulate for? What phase should they be set for stimulation? Which brain regions should be separately/simultaneously targeted? And how long should the stimulation duration be?

## 4. Cognition in Rehabilitation

Two factors contribute to progress in rehabilitation: (1) motivation and (2) fatigue. Both motivation and fatigue are cognitive processes that can either be fleeting or persistent sensations/perceptions experienced by an individual [43] (Figure 2).

To accurately reflect epochs of motivation or fatigue using methods such as questionnaires or self-reports can be challenging because the subject may or may not be voluntarily aware of these subliminal states of mind. Hence, researchers relied on correlates that could provide an insight in quantifying cognitive components such as motivation and fatigue. To understand the brain’s motivational network, studies have shown that associated brain regions would increase the saliency of a relevant stimulus and become highly attuned in paying attention to it [44]. Changes in attentional activity potentially modulate working memory representations that would, in turn, be bound temporally as a discrete component in a memory episode. Prospectively, investigating the domains of working memory and attention could act as “proxies” on measuring the degree of motivation in subjects during an assigned task (Figure 2).

The following subsections aim to discuss some of the recent findings in the domains of working memory, attention, and fatigue, brain oscillation modulation from tACS delivery, and to suggest neurophysiological explanations to rationalize findings from tACS manipulations in the relevant literature (chosen examples out of a table of tACS literature reviewed). Finally, the paper will conclude with a framework integrating these domain-specific tasks in a MI-tACS protocol to be proposed for a more holistic patient rehabilitation in the future.

### 4.1. Working Memory-Related tACS Studies

Working memory can be defined as the ability to store information temporarily to guide one’s behavior. MI tasks are often instructed to subjects as visual cues or representations that require storage in working memory before reactivating that stored information after an extremely brief interval without motor execution. Since no actual motor movements are involved in the task, participants will need to be consciously engaged to not only comprehend the given task instructions but also to execute motor imagery when prompted as well. Hence, it can be said that working memory tasks can be good proxies to interrogate an individual’s motivation towards MI.

Analogous to the human working memory task, Siegle and Wilson (2014) revealed that the theta oscillation cycle generated in the navigating mouse’s hippocampal CA1 region is modulated by inputs from the entorhinal cortex (located in the medial temporal lobe) and the hippocampal CA3 region, which coincide at theta cycle’s trough and peak, respectively [45]. Consequently, inhibiting entorhinal cortex inputs at the troughs inhibited the ability to encode spatial information, while inhibiting hippocampal CA3 inputs at the peaks inhibited information retrieval (i.e., recall). This is one of the first keynote studies that demonstrated different functional properties of the brain (i.e., information encoding and retrieval) associated with specific oscillatory phases of a particular brain wave.

This finding from the rodent is in line with work by Alekseichuk et al. (2016) [46] (Figure 3, in which they showed that enhanced spatial working memory was observed when theta frequency is manipulated at its peak of the cycle, likely a memory retrieval facilitation. Specifically, they delivered superposed high frequency gamma bursts with theta oscillation via tACS over the left parietal region, where only a cross-frequency coupling of theta and 80–100 Hz gamma bursts would elicit a modulation of enhanced spatial working memory. The similar group of authors then attempted to extend this work by delivering cross-frequency coupling of theta and 80–100 Hz gamma bursts at the peaks and troughs of theta, respectively, over the left temporal lobe region during the encoding phase of the given task to investigate long-term verbal associative memory [47]. Although gamma bursts superposed at theta’s troughs delivered during encoding impaired performance, gamma bursts superposed at theta’s peaks showed no significant effect. It is possible that memory consolidation for long-term storage is likely projected and redistributed to higher order cortices; therefore, it would not be reactivated effectively for retrieval at the temporal lobe that they had stimulated.

The majority of research conducted in this domain advocates that theta-gamma coupling in the parietal region of the brain showed positive impacts on working memory, and that the parietal region is implicated in working memory function [48,49,50]. There is also evidence of tACS resulted in a more significant effect on working memory compared to tDCS for the same task [51,52] (Table 2). All in all, the common strategy seems to target the left parietal region to modulate theta and gamma coupling, while the theta frequency is shown to process encoding and retrieval phases of memory while being a “temporary store” for gamma frequency to append mnemonic items of a task to the frequency-coupled oscillation for retrieval.

Recent clinical studies have shown that delivery of online tACS resulted in working memory improvements, along with other cognitive improvements such as attention, reaction time, and emotional processing in schizophrenia patients [53]. Though no EEG analysis was performed, they testify the potential and feasibility of tACS in enhancing performance of working memory, which can be extended to other patient groups suffering from such cognitive impairment, including stroke.

**Table 2 brainsci-13-01584-t002:** Working Memory-related Studies.

Working Memory-Related Studies				
Study	Subjects and Design	Experimental Task	Stimulation Montage and Parameters	Outcome
Alekseichuk et al. (2016) [46]	47 healthy adults (22 males, 25 females)	2-back visual-spatial match-to-sample test	Active electrodes: central electrode over AF3 (10–10 system), the other 4 were equally spaced at 6 cm from AF3. 1 mA (peak-to-peak), 10 min (including 10 s ramp-up and ramp-down periods), 0.6 mA peak-to-baseline. Sham stimulation 1:30 s and then turned off. Sham stimulation 2:80 Hz at a lower intensity (0.2 mA peak-to-baseline).	θ and θ-γ frequency coupling improve working memory performance. High γ power (80 Hz γ bursts) coupled over the peak, but not the trough, of θ improves working memory. Optimal γ frequencies for improved performance are in the range of 80 to 100 Hz.
de Lara et al. (2018) [47]	72 healthy adults (36 males, 36 females). Between-group design.	Paired-associative learning task using word-pairs	Active electrode at T7 and return electrodes at FPz and T8 (10–20 system). 1 mA (peak-to-baseline), 10 min (including a 10 s ramp-up and 10 s ramp-down). Sham stimulation: 10 s ramp-up, current delivered for 30 s, 10 s ramp-down.	γ bursts coupled to the troughs of θ-tACS resulted in behavioral impairment in memory performance. γ bursts coupled to the peaks of θ-tACS or superimposed over the whole θ-tACS waveform resulted in no significant behavioral effects.
Thompson et al. (2021) [48]	51 healthy adults (21 males, 30 females). Within-subject design.	Visual retro-cue task	Active electrodes over P3 and P4 (10–20 system). 1.5 mA (20 s ramp-up and ramp-down 20 s), 20 min. Sham stimulation: 20 s.	Parietal γ-tACS resulted in significant improvement in recall precision only during invalid cue trials and only for high-baseline performers (assessed using a multi-level mixed model).
Tseng et al. (2016) [49]	20 healthy adults (12 males, 8 females). Within-subject design.	Change detection task	Active electrodes at CP1 and T5 (10–20 system). 1.5 mA (peak-to-peak), 20 min (switched off in last 2 blocks). Sham stimulation: only for 30 s.	γ-tACS improved task performance only for shape-color binding trials in low-performers (based on d’ index from hit and false alarm rates). Improved performance lasted throughout the last 2 blocks (final 20 min offline session).
Lang et al. (2019) [51]	59 healthy adults. Between-group design.	Visual associative memory task (designed in-house): Face and Scene Task (FAST)	Active electrodes at FP1, P2, P3, PO7, P10; anode at P10. 2 mA (peak-to-baseline), 10 min (30 s ramp-up and ramp-down). Sham stimulation: 30 s ramp-up to 2 mA, 30 s ramp-down to 0.06 mA for 9 min.	θ-tACS improved visual associative memory performance (based on correct hits and # errors). θ-tACS facilitated reduction in false memory and forgetting (based on # errors). Results suggest that tACS is more effective than tDCS.
Röhner et al. (2018) [52]	30 healthy adults (15 males, 15 females). Within-subject design.	2-back visual letter task	Active electrodes placed over F3 and P3 (10–20 system). 1 mA, 15 min (including a 15 s ramp-up and 15 s ramp-down). Sham stimulation: 1 min tDCS.	Offline effect of θ-tACS on RTs showed significant improvement in performance (not observed in anodal tDCS condition).
Abellaneda-Pérez et al. (2020) [54]	44 healthy adults (22 males, 20 females). Between-group design.	Verbal n-back task	Active electrode over F3 and return electrode over FP2 (10–10 system). 2 mA (peak-to-peak), 20 min (15 s ramp-up and ramp-down). Sham stimulation: terminated after 30 s of delivery.	θ-tACS effects suggested to be driven by online brain activity changes during stimulation, but not post-stimulation. Enhanced brain activity was found in θ-tACS group in frontal, parietal and thalamic areas during lowest working memory load, and in right frontal areas during highest working memory load.
Meng et al. (2021) [55]	20 healthy adults (8 males, 12 females). Within-subject design.	Visual associative memory task (designed in-house): Face and Scene Task (FAST), perceptual recognition test	Active electrode over P3 (10–20 system). 2 mA (peak-to-peak), 15 min (30 s ramp-up and 30 s ramp-down). Sham stimulation: 50 s.	θ-tACS impaired associative memory performance (based on d’ from hit and false alarm rates). No significant difference between θ-tACS and sham groups for perceptual recognition test.
Pahor and Jaušovec (2018) [56]	72 healthy adults (females). Within-subject design.	Change detection tasks, n-back tasks	Pair combination of active electrodes placement over F3, F4, P3, P4 (10–20 system). 1.25 mA to 2 mA stepwise increment over 30 s, 15 min. Sham stimulation: 1 min.	θ-tACS modulates for significant changes in post-stim resting-state EEG amplitude relative to baseline. θ-tACS facilitated small improvements in performance only for certain n-back tasks. θ-tACS resulted in significant ERP amplitude and latency changes in n-back tasks compared to sham.
Nomura et al. (2019) [57]	36 healthy adults (8 males, 28 females). Between-group design	Visual word recognition task	Active electrode over F3 (10–20 system). 750 uA (peak-to-baseline), 15 min, 100 cycles fade-in and fade-out. Sham stimulation: 10 s.	γ-tACS applied over the left prefrontal cortex enhances episodic memory (i.e., long term recall) response accuracy without affecting reaction time.

### 4.2. Attention-Related tACS Studies

Attention can be defined by two sub-processes: covert and overt attention. Covert attention is said to be sustained by the brain’s attentional network driven by goal-directed intentions without physically attending to a stimulus, while overt attention is debated to be the by-product of covert attention in which a saccade is made to fixate on a stimulus with a goal-directed intention [58]. When attending to a stimulus, visual information is first encoded by the primary visual area (V1) and is then transmitted via two main streams of processing pathways: the dorsal and ventral pathways [59], (Figure 4). While the former directs sensory information flow to the posterior parietal cortex, a region implicated in visuospatial perception and planning, the latter directs information flow to the inferior temporal cortex, a region implicated in stimulus recognition. Primate studies showed that an enhancement in neuronal firing rates recorded in the visual attention network does not truly reflect attentional effects [60]. Fixating on a stimulus and having it in the receptive field of neurons to trigger firing does not imply that one is actively processing the stimulus. Instead, both increases and decreases in firing rates of the neuronal populations mediate the overall stimulus encoding process to result in conscious attention. This synchronization of population-wide firing rates across cortices in the attention network supports the notion that modulations in respective oscillatory frequencies could be picked up by EEG recordings. Since motivation can be deemed as a goal-directed control of attention, outcomes measured from attentional tasks can also be used to validate an individual’s motivation.

The majority of research conducted in this domain using tACS have pinpointed four different frequency bands that underlie attention: theta, alpha, beta, and gamma oscillation frequencies (Table 3). Kasten et al. (2020) and Schuhmann et al. (2019) showed an effect only on top-down attention that is specific to the delivery of tACS in the alpha or gamma frequency to the occipital cortex and posterior parietal cortex, respectively [61,62]. However, when the same alpha- and gamma-tACS were applied to the inferior parietal lobe, modulatory effects was differentiated in bottom-up and top-down attention modulation, respectively [63]. An EEG study in elderly participants by Fahimi et al. (2018) further supported that alpha and gamma frequency bands are implicated in transmitting attentional information [64].

Studies have also documented the targeting of either the dorsolateral lateral prefrontal cortex (DLPFC) or medial prefrontal cortex (mPFC), since PFC is a region implicated in executive function. While Lehr et al. (2019), Klírová et al. (2021), and Rostami et al. (2020) [65,66,67] stimulated the PFC in the theta frequency, Moliadze et al. (2019) [68] had done so in the alpha frequency instead. Nonetheless, a modulatory effect on alpha phase synchrony and power was observed even with theta-frequency tACS [67]. Thus, it seems to suggest a role of frequency coupling between alpha and gamma in the brain’s attentional network as a gating mechanism for inhibitory control [69] in order to guide attentional shifts and encoding of relevant information. On the other hand, theta-beta coupling seems to serve the role of temporary information storage while further down-playing the saliency of irrelevant information, suggesting a supporting role to sustain and modulate activity from alpha-gamma frequency coupling. It may not be surprising that mechanisms of the attention network subserve the working memory function since a goal-oriented attention shift for encoding would imply its relevance to be stored in working memory (Figure 2). Hence, it can be postulated that cortices processing sensory information such as the occipital and posterior parietal cortices are likely to be modulated by higher frequencies in the spectra of brain oscillations (e.g., alpha and gamma), and cortices processing information in higher orders of cognition such as the prefrontal cortex tend to be modulated by lower frequencies (e.g., theta) in a concerted effort that temporally binds attention and working memory.

Clinical studies on attentional deficits are conducted in patients with attention-deficit/hyperactivity disorder (ADHD). To date, there is only one reported study that utilized tACS in an ADHD patient group using tACS, and this study found a resulting decrease in error rates [70]. Although still in its infancy, it is promising evidence to validate tACS use in patients, with likely extensions to other patient groups in future.

**Table 3 brainsci-13-01584-t003:** Attention-related Studies.

Attention-Related Studies				
Study	Subjects and Design	Experimental Task	Stimulation Montage and Parameters	Outcome
Kasten et al. (2020) [61]	20 healthy adults (10 males, 10 females). Crossover within-subject randomized design.	Spatial cue task	2 pairs of electrodes at O1-P3 and O2-P4 (10–20 system). 1 mA (peak-to-baseline), ~8 min per block.	tACS modulates RTs only in endogenous attention. Right occipital α-tACS increases RTs in both valid and invalid cue trials. γ-tACS contralateral-to-cue modulated RTs more significantly than α-tACS ipsilateral-to-cue as compared to that for γ-tACS: observation only for invalid cue trials.
Schuhmann et al. (2019) [62]	36 healthy adults (18 males, 18 females). Within-subject design.	Spatial cue task and Stimulus detection task	P3 (10–20 system). 1 mA (peak-to-peak), >40 min for all tasks. Sham stimulation: 100 cycles, ramped up and down immediately.	Larger leftward bias in RTs during endogenous attention task in tACS group.
Hopfinger et al. (2016) [63]	23 healthy adults (9 males, 14 females). Within-subject design.	Spatial cue task	P6 and Cz (10–20 system). 1 mA (peak-to-baseline). Sham stimulation: 30 s (4 s ramp-up, maintained for 22 s, 4 s ramp-down).	Significantly slower RTs in invalid/uncued exogenous trials with α-tACS. Significantly faster RTs in invalid endogenous trials with γ-tACS.
Klírová et al. (2021) [65]	20 healthy adults (10 males, 10 females). Crossover design.	Simon task, Stop Signal task, Conner’s Continuous Performance Test 3rd edition (CPT III), Stroop test	FCz and Pz. 1 mA (peak-to-baseline), 30 min (5 s ramp-up and 5 s ramp-down). Sham stimulation: 5 s ramp-up, 30 s, 29 min 30 s rest, 5 s ramp-down.	Only significant effect observed in the Stroop color-word test and Stroop interference scores, with better performance in the individualized stimulation group compared to that of the non-individualized group. However, neither group differed significantly in performance when compared to sham.
Lehr et al. (2019) [66]	22 healthy adults (6 males, 16 females). Within-subject design.	Stroop color-word task	AF3, 4 return electrodes at F5, F2, Fp2, AF7 (10–10 system). 1 mA (peak-to-baseline), 20 min (including 10 s ramp-up and ramp-down). Sham stimulation: 30 s; at the beginning and end of the task.	θ-tACS reduced Stroop effect only in trials preceded by congruent trials.
Rostami et al. (2020) [67]	13 healthy adults (7 males, 6 females). Within-subject design.	Rapid visual information processing (RVIP) task from CANTAB	Fpz (10–20 system). 1 mA (peak-to-peak), 20 min, (10 s ramp-up and down). Sham parameters: 30 s.	6 Hz θ-tACS increased frontal-midline theta and resulted in significant changes in RVIP scores. Faster RTs for correct responses with 6 Hz θ-tACS. EEG power analysis showed changes in theta PSD in frontal, central, and temporal regions in right hemisphere.
Moliadze et al. (2019) [68]	24 healthy adults (12 males, 12 females). Within-subject design.	Phonological task (words)	1 electrode each located between F1, F5, FC3 (left) and F2, F6, FC4 (right). 1 mA, 20 min (15 s ramp-up and down). Sham stimulation: 15 s ramp-up, 30 s at 1 mA, 15 s ramp-down.	10 Hz α-tACS significantly facilitates phonological word decision RTs. 10 Hz α-tACS significantly increases task-related theta power during phonological decisions.
Hutchinson et al. (2020) [71]	71 healthy adults (34 males, 37 females). Between-group design.	Inattentional Blindness (IB) task developed by Pitts et al. (2012)	Oz and Cz. Current intensity customized to subject’s level of comfort or subjects reported phosphenes. Sham stimulation: a mild current of 30 s.	α-tACS group: 87% inattentionally blind; θ-tACS group: 45.8% inattentionally blind; Sham group: 50% inattentionally blind; α-tACS less perceptive of target stimulus than those with θ-tACS or sham applied.
van Schouwenburg et al. (2017) [72]	37 healthy adults. Between-group design.	Spatial cue task	F4 and P4. 1 mA (peak-to-baseline), 5 min (15 s ramp up and down). Sham stimulation: immediate ramp down over 15 s.	Sham group showed significant attention bias (faster RTs to targets) in the right hemifield compared to the left hemifield. Sham group showed significant lateralization in frontoposterior alpha coherence that was not present in the α-tACS group. In α-tACS group, they found a relative increase in right hemispheric coherence (relative to the left) and an attentional shift towards the left hemifield.

### 4.3. Fatigue-Related tACS Studies

Fatigue can be subdivided into two categories: active and passive fatigue. Active fatigue stems from both cognitive and motor overload of extended task-related events, while passive fatigue stems from cognitive and motor underload over extended periods of time (Figure 5). The consequences of active fatigue are often seen in a decline in physical or cognitive performance with extended reaction times in tasks. In stroke patients executing MI for prolonged periods, they too become actively fatigued from mental tasks. On the other hand, in passive fatigue (also known as driver fatigue [73]), an individual could potentially continue with the task, performing at substantial levels, without being consciously aware of fatigue setting in. For stroke patients, it could potentially reduce rehabilitation efficacy because they would be denying the brain’s motor and executive function networks the need to rest and consolidate learned processes.

There are two existing methods to measure fatigue: subjective qualification via surveys and objective quantification via EEG readouts such as a decrease in beta activity [34], and an increase in theta activity [74]. Changes in alpha activity are also considered as a candidate marker of fatigue detection but studies on this have given rise to conflicting results as to whether it increases or decreases during fatigue [75,76].

Work on the domain of fatigue has mostly adopted the strategy of enhancing attention to counter fatigue by stimulating brain regions implicated in attention, such as gamma frequency tACS in the visual cortex [77]. However, their findings only show an improvement in reaction times but not performance accuracy in a timely manner; this therefore implies that targeting the attention network may not ameliorate fatigue per se nor all of its implications (Figure 2).

Recently, effective computational algorithms have been developed for semi-supervised learning using labeled EEG data from an attended task before testing it on unlabeled EEG data [78], (Table 4). From there, the algorithm could identify EEG fatigue signatures of an individual. Apart from detection of fatigue EEG signals, a subsequent study by Foong et al. (2020) validated that lower beta band power in the frontal and central regions are correlates for MI performance in the presence of fatigue [79]. These developments and findings provide evidence that fatigue can be objectively identified even without a conscious awareness of it.

Fatigue often presents itself as a comorbid condition in various neuromuscular and neurocognitive disorders: stroke, depression, Parkinson’s Disease, multiple sclerosis etc. At present, it is shown that tDCS is effective at alleviating fatigue per se in patients with multiple sclerosis [80,81]. Nonetheless, with current fatigue detection algorithms, it seems promising that real-time intervention or treatment for alleviating fatigue using tACS will happen in the near future.

**Table 4 brainsci-13-01584-t004:** Fatigue-related Studies.

Fatigue-Related Studies (Non-tACS)				
Study	Subjects and Design	Experimental Task	Recording Montage	Outcome
Huang et al. (2016) [75]	12 healthy adults (7 males, 5 females).	Virtual Reality-based highway driving: event-related lane-departure/deviation task	32-channel EEG recording electrodes (10–20 system).	Increased theta activity in frontal midline and occipital areas. Average RTs of epochs with auditory warning maintained at 1.15 times the mean RT. Significantly slower RTs in trials in which α power exceeded warning threshold but was not given an auditory warning. Occipital EEG power spectra in θ and α bands decreased rapidly with warnings.
Foong et al. (2019) [78]	29 healthy adults	Driving simulation task	EEG electrodes at FP1, FP2, TP9, TP10 (10–20 system).	Semi-supervised learning using labeled attentive data to predict and identify passive fatigue from unlabeled data.
Zhang et al. (2021) [82]	48 healthy adults (24 males, 24 females).	Detection response task (DRT) in driving simulator	64-channel EEG recording electrodes (10–20 system)	DRT performance declines at 40 min (based on RTs and response accuracy). α power was significantly higher in the automated driving group as opposed to the manual driving group; indicative of passive fatigue.

## 5. Discussion

When designing for a holistic and personalized treatment paradigm, some may define “personalized” as adapting the delivered tACS to an individual’s changing endogenous frequency during the task or having an individualized electrode montage for tACS delivery. However, Stecher et al. (2021) have shown that there is no significant increase in post stimulation power in the adaptive frequency stimulation group compared to the fixed frequency stimulation group [83]. Similarly, Klírová et al. (2021) only managed to demonstrate positive effects from using an individualized electrode montage in tasks that involved verbal components [65]. No significant benefit was observed in non-verbal tasks. It was also reported that entrainment effects result from tACS.

A personalized treatment paradigm should take factors such as working memory, attention, and fatigue into account with real-time monitoring in a closed-looped system with an MI-tACS protocol (Figure 2). The ultimate goal would be to improve cognitive domains apart from motor recovery/learning, but the first steps to take with relatively new technologies such as tACS would be to focus on one domain first (e.g., MI) while detecting/monitoring changes via EEG in other domains (e.g., working memory and fatigue) concurrently (Figure 6). MI classification techniques are seeing increasingly advanced algorithms developed to increase both accuracy in classification performance and efficiency in computational processes [84,85,86], aspects wherein further developments will unfold swiftly in the foreseeable future. More studies are required to define individualized parameters such as electrode placements for stimulation, stimulation frequency and amplitude, the appropriate experimental task type and its level of difficulty should be carefully designed for every patient. Once achieved, the MI-tACS protocol can then be improved by interleaving sessions of working memory and attentional tasks in a hybrid design to enhance both motor and cognitive outcomes while monitoring the patient’s fatigue cycles. In doing so, one would be able to fulfill not only the “dose (i.e., parameters) and frequency (i.e., fatigue-dependent schedule)” requirements from a therapist’s standpoint but also achieve a holistic rehabilitation paradigm that covers both motor and non-motor aspects for the patients as well.

## 6. Conclusions

Transcranial electrical stimulation has transcended over a decade of research to date, and we are only beginning to understand some of the implications it has on modulating brain oscillations, facilitating functional organization and how it might facilitate patient rehabilitation processes. However, we are only at the “tip of the iceberg” in comparison to the degree of complexity that is associated with understanding how such non-invasive modalities alter underlying mechanisms of the brain. One thing we can be certain about is that the brain’s functional networks never operate in a one-to-one manner i.e., one input to produce one output, or one type of affector paired to only one type of effector. Since the brain is a dynamic interconnected global network of many sub-networks, its many-to-one and one-to-many functional relationships will need to be considered in order to realize complex behaviors including that of patient rehabilitation.

Moving forward, the paper wants to emphasize the importance of a multifaceted integration of cognitive domains into motor imagery and tES related tasks to encourage a more holistic rehabilitation protocol design that can be customized in a patient-centric manner for the next breakthrough in the near future.

## Figures and Tables

**Figure 1 brainsci-13-01584-f001:**
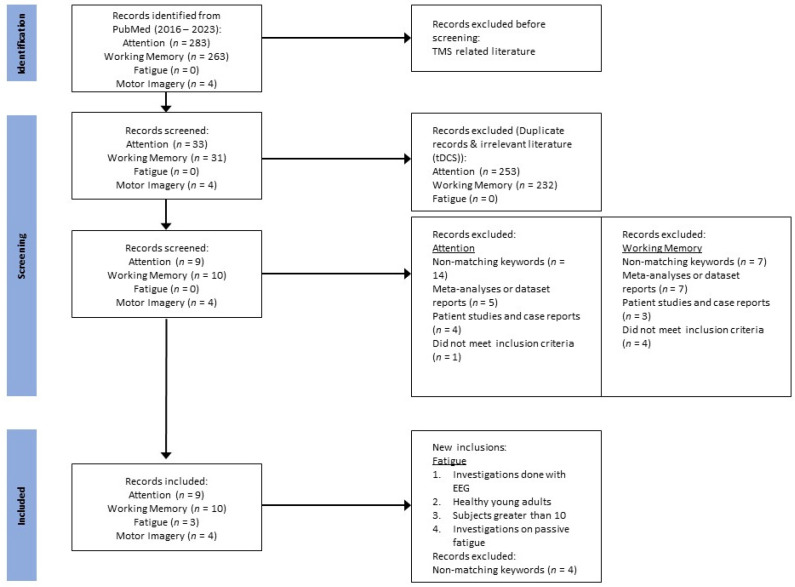
Literature selection procedure with ‘*n*’ indicating the number of papers/articles of the respective categories.

**Figure 2 brainsci-13-01584-f002:**
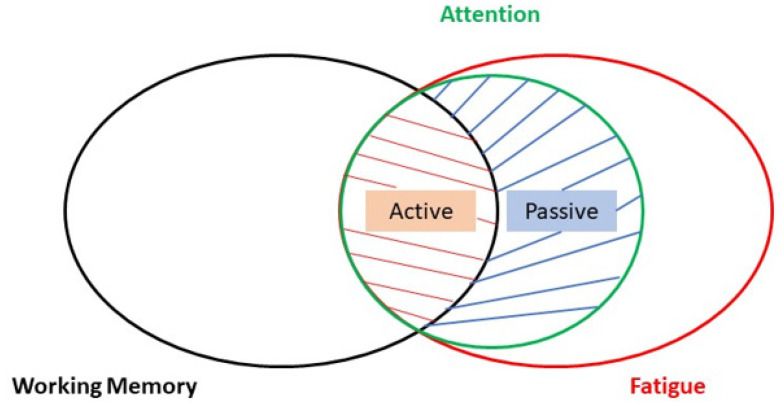
A Venn diagram illustrating the cognitive implications amongst three domains: working memory, fatigue, and attention. Active fatigue affects attention and working memory; passive fatigue mainly impacts attention.

**Figure 3 brainsci-13-01584-f003:**
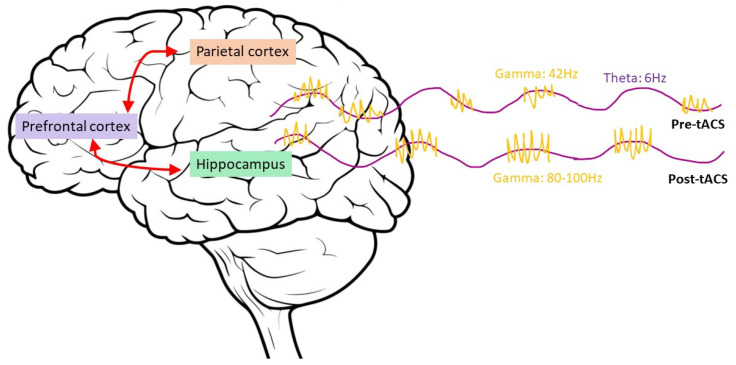
Illustrating the network of plausible brain regions implicated in working memory, and that tACS delivery of high frequency gamma coupled to theta peaks facilitates working memory within this network.

**Figure 4 brainsci-13-01584-f004:**
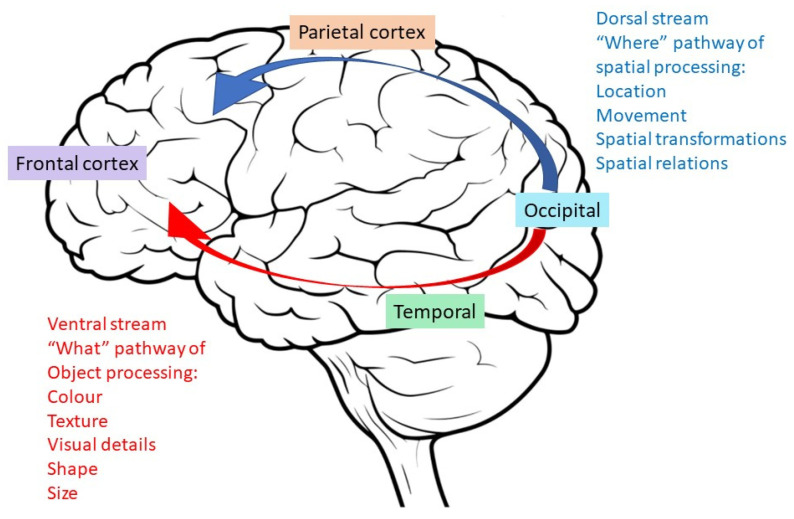
An illustration depicting the two main streams of visual processing—the dorsal “where” and ventral “what” streams—as well as the cortices implicated in the respective processes that are directed by goal-directed attention.

**Figure 5 brainsci-13-01584-f005:**
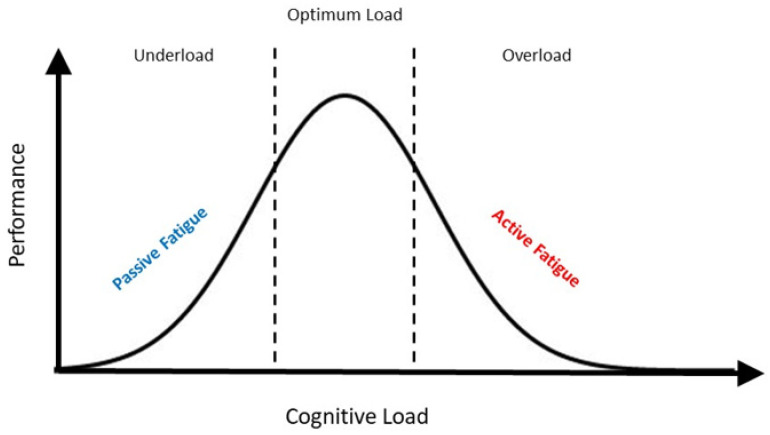
A diagram illustrating the relationship curve between performance and cognitive load of an individual.

**Figure 6 brainsci-13-01584-f006:**
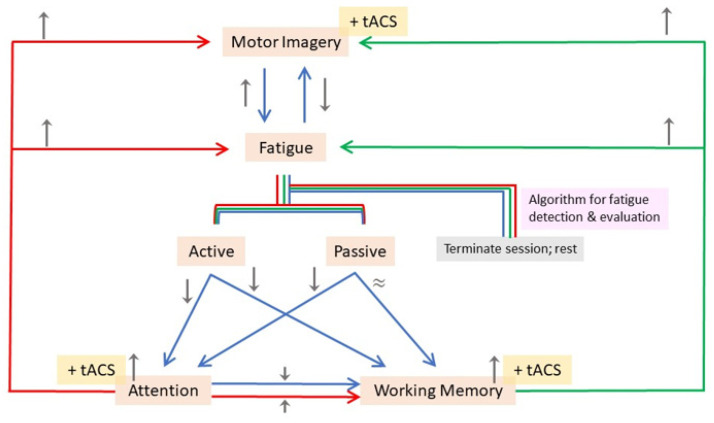
Proposed framework of MI-tACS, working memory, attention, and fatigue for rehabilitation. Fatigue detection determines therapy duration. Color-coded arrows: implications of tACS when utilized in each domain (MI: blue; Attention: red; Working Memory: green). Grey arrows: up-and down-regulation accordingly. “≈” symbol: no substantial impact. Tri-color line segments: shared routes in the flowchart. The schematic begins with the execution of motor imagery in the presence of tACS to facilitate MI. The flow of the schematic (i.e., the inter-relationship between the components of MI, Fatigue (Active and Passive), Attention and Working Memory) is indicated by the blue arrows. Subsequently, the schematic shows how the application of tACS facilitates the components of Attention and Working Memory, which in turn affects MI and Fatigue correspondingly. Concurrently, monitoring of Fatigue levels are monitored via an algorithm for fatigue detection and evaluation.

**Table 1 brainsci-13-01584-t001:** Motor Imagery-related Studies.

Motor Imagery-Related Studies				
Study	Subjects and Design	Experimental Task	Recording Montage	Outcome
Xie et al. (2021) [30]	15 male healthy adults.	Hand-grasping MI task.	16-channel EEG recording electrodes (10–20 system). Anode at Cz.	Enhanced ERD of *μ* and *β* rhythms in left-hand MI task. Both average classification accuracy of tACS (88.19%) and tDCS (89.93%) groups improved significantly compared to pre-a nd sham groups.
Brinkman et al. (2016) [40]	38 healthy adults (16 males, 22 females). Within-subject design.	MI task of grasping a tilted cylinder with either left or right hand	Stimulating electrodes over C3 and C4, reference at Pz (10–20 system).	A and *β* band oscillations have dissociable effects on movement selection. A band stimulation resulted in faster responses.
Naros and Gharabaghi (2016) [41]	20 severely affected chronic stroke patients. Parallel group design.	Kinesthetic MI programmed into a Brain-robot interface (BRI)	32-channel EEG recording with 1 stimulating electrode on contralesional brain region (10–20 system).	No sustained offline effects of *β*-tACS. No evidence of *β*-tACS facilitating motor skill acquisition or motor consolidation. *β*-tACS shown to stabilize intrinsic *β*-fluctuation to improve BRI performance. Stimulation paradigm did not influence MI-related *β*-ERD.
Zhang et al. (2023) [42]	36 healthy adults. Randomized control design.	Hand-grasping MI task; Letter-writing MI task	Stimulating electrodes at P4 and F4, reference at Cz (10–10 system).	*μ* rhythm ERD and classification accuracy improved after anti-phase tACS. Anti-phase tACS caused ERD between frontoparietal network regions in letter-writing MI task. No beneficial effects of anti-phase tACS in the hand-grasping MI task.

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
