# Peer review of "A Review on Motor Imagery with Transcranial Alternating Current Stimulation: Bridging Motor and Cognitive Welfare for Patient Rehabilitation"

_brainsci, 2023, doi:10.3390/brainsci13111584_

Round 1

Reviewer 1 Report

Comments and Suggestions for Authors

The paper is interesting and within the scope of journal. Hovewer, paper fails to meet the usual quality level of journal papers. Authors are advised to consider the following modifications:

* Please reconsider abstract, it is not clear whether paper presents novel results or solely review of reported results. It might be advisable to reconsider paper title to reflect review nature of the paper.

* Please adapt last paragraph of the Introduction to better stress novelty of the papaer beyond review of published results. What does it mean to briefly discuss? Are some conclusions devised?

* Please reconsider caption of Figure 1. It appears that not only the procedure is presented, but also some results (number of the selected papers), or not?

* Title of the section 3 should be rewritten not to include only abbreviations.

* Correct numbering of the figures, Figure 1 appears twice in the paper.

* Tables break across to many pages and are thus difficult to read. Consider editing to compact them.

* Caption of the figure 4 appears detached from the figure itself.

* Conclusions section should not contain figures especially not at the end. If the proposed framework is the paper novelty, it should be introduced earlier in the paper.

* The structure of the paper has to be reconsidered, since this is the weakest point of the work. Paper levitates between review paper and some kind of research paper. Discussion section should probably be introduced where majority of the contents from conclusion should be moved, while conclusions should be rewritten to present some findings concisely.

Comments on the Quality of English Language

Only minor editing of English is required and final proofreading, but only after final significant modifications of the paper.

Author Response

Dear Reviewer 1,

The authors would like to express their sincere gratitude for taking time and patience in thoroughly reviewing the manuscript and providing constructive feedback to bolster its content and quality. 

Please see the attachment for the authors' response to the posted comments and the corresponding changes made to the manuscript.

Thank you very much once again.

Sincere regards,

Rosary Lim

Reviewer 2 Report

Comments and Suggestions for Authors

In this review article, the authors present detailed discussions of the existing articles focusing on the cognitive domains of working memory, attention, and fatigue in relation to various types of transcranial electrical stimulation (tES) techniques. The title, "Motor imagery with transcranial alternating current stimulation: bridging motor and cognitive welfare for patient rehabilitation," sets an expectation of a primary focus on tACS techniques to improve motor imagery tasks for rehabilitation. However, there is no dedicated summary table for motor imagery-rated studies similar to the ones presented in Tables 1-3. While the title emphasizes the importance of motor imagery,  the content predominantly discusses three cognitive areas.  The discussion on "Bridging motor and cognitive welfare" is briefly described in the conclusion. Therefore, I recommend that the authors consider adding one more section to elucidate existing tACS techniques for motor imagery and provide a deeper exploration of how the integration of motor and cognitive functions can enhance patient rehabilitation would substantively enrich the manuscript. 

Author Response

Dear Reviewer 2,

The authors would like to express their sincere gratitude for taking time and patience in thoroughly reviewing the manuscript and providing constructive feedback to bolster its content and quality. 

Please see the attachment for the authors' response to the posted comments and the corresponding changes made to the manuscript.

Thank you very much once again.

Sincere regards,

Rosary Lim

Round 2

Reviewer 1 Report

Comments and Suggestions for Authors

All my concerns have been properly considered. I believe that paper is much improved. I have the single remaining suggestion:

* Please rephrase novel caption for Figure 1 to make it single sentence, for exapmple:

Figure 1. Literature selection procedure with indicated number of papers of the respective categories marked with n.

or similar.

Comments on the Quality of English Language

Only final proofreading needed.

Author Response

Dear Reviewer 1,

The authors would like to offer their sincere gratitude in taking time to provide suggestions to further refine the manuscript towards publication.

Please see the attachment for the responses towards the comment given.

Once again, thank you for the kind patience and time taken for reviewing the manuscript.

Sincere gratitude,

Rosary Lim

Reviewer 2 Report

Comments and Suggestions for Authors

The authors have adequately addressed my concerns about motor imagery-related tACS studies. Therefore, I recommend that this revised manuscript be accepted for publication in this journal. 

Author Response

Dear Reviewer 2,

The authors would like to offer their sincere gratitude for your time and patience in offering constructive suggestions towards improving the quality of the manuscript; and are honored to have received your approval in acknowledging it for publication.

Once again, thank you very much for your kind words and time in our manuscript.

Sincere gratitude,

Rosary Lim